# A Boundary-Enhanced Liver Segmentation Network for Multi-Phase CT Images with Unsupervised Domain Adaptation

**DOI:** 10.3390/bioengineering10080899

**Published:** 2023-07-28

**Authors:** Swathi Ananda, Rahul Kumar Jain, Yinhao Li, Yutaro Iwamoto, Xian-Hua Han, Shuzo Kanasaki, Hongjie Hu, Yen-Wei Chen

**Affiliations:** 1Graduate School of Information Science and Engineering, Ritsumeikan University, Kusatsu-shi 525-0058, Japan; gr0502if@ed.ritsumei.ac.jp (S.A.); rahulkumarjain16@gmail.com (R.K.J.); yin-li@fc.ritsumei.ac.jp (Y.L.); 2Faculty of Information and Communication Engineering, Osaka Electro-Communication University, Neyagawa-shi 572-0833, Japan; yiwamoto@osakac.ac.jp; 3Artificial Intelligence Research Center, Yamaguchi University, Yamaguchi-shi 753-8511, Japan; hanxhua@yamaguchi-u.ac.jp; 4Koseikai Takeda Hospital, Kyoto-shi 600-8558, Japan; shuzo65@gmail.com; 5Department of Radiology Sir Run Run Shaw, Zhejiang University, Hangzhou 310016, China; hongjiehu@zju.edu.cn

**Keywords:** multi-phase CT image, liver segmentation, deep learning, unsupervised domain adaptation, boundary enhancement

## Abstract

Multi-phase computed tomography (CT) images have gained significant popularity in the diagnosis of hepatic disease. There are several challenges in the liver segmentation of multi-phase CT images. (1) Annotation: due to the distinct contrast enhancements observed in different phases (i.e., each phase is considered a different domain), annotating all phase images in multi-phase CT images for liver or tumor segmentation is a task that consumes substantial time and labor resources. (2) Poor contrast: some phase images may have poor contrast, making it difficult to distinguish the liver boundary. In this paper, we propose a boundary-enhanced liver segmentation network for multi-phase CT images with unsupervised domain adaptation. The first contribution is that we propose DD-UDA, a dual discriminator-based unsupervised domain adaptation, for liver segmentation on multi-phase images without multi-phase annotations, effectively tackling the annotation problem. To improve accuracy by reducing distribution differences between the source and target domains, we perform domain adaptation at two levels by employing two discriminators, one at the feature level and the other at the output level. The second contribution is that we introduce an additional boundary-enhanced decoder to the encoder–decoder backbone segmentation network to effectively recognize the boundary region, thereby addressing the problem of poor contrast. In our study, we employ the public LiTS dataset as the source domain and our private MPCT-FLLs dataset as the target domain. The experimental findings validate the efficacy of our proposed methods, producing substantially improved results when tested on each phase of the multi-phase CT image even without the multi-phase annotations. As evaluated on the MPCT-FLLs dataset, the existing baseline (UDA) method achieved IoU scores of 0.785, 0.796, and 0.772 for the PV, ART, and NC phases, respectively, while our proposed approach exhibited superior performance, surpassing both the baseline and other state-of-the-art methods. Notably, our method achieved remarkable IoU scores of 0.823, 0.811, and 0.800 for the PV, ART, and NC phases, respectively, emphasizing its effectiveness in achieving accurate image segmentation.

## 1. Introduction

The death rate due to liver diseases caused by cancer, infection, excessive alcohol intake, and abnormal immune system function has been rising in recent years [1]. Multi-phase CT images are widely used to diagnose disease more accurately [2]. Multi-phase CT scans the human body at different time periods after injecting a contrast medium. The various phases of multi-phase CT images consist of the non-contrast (NC) phase, which is a native image captured prior to the injection of contrast [2], the arterial (ART) phase, which is an image acquired 15–35 seconds after contrast injection to enable visualization of the arteries, and the portal venous (PV) phase, which is an image obtained between 60–80 seconds after injection to visualize the portal venous. Typical multi-phase CT images with different liver lesions are shown in Figure 1. By analyzing the enhancement patterns of the multi-phase CT images, radiologists can make a better accurate diagnosis than the standard low-contrast single-phase CT [3]. Although multi-phase CT is convenient, the number of images that need to be checked increases, which in turn increases the burden on radiologists. As a result, demand for computer-aided diagnosis (CAD) systems is increasing. In this paper, we address automatic liver segmentation as an essential CAD system.

Over the past few years, the application of deep convolutional neural networks (CNNs) has shown remarkable achievements, particularly in detection and segmentation tasks, including medical imaging [4,5], including liver segmentation in CT images [6,7,8]. However, there are several challenges in the segmentation of multi-phase CT liver images. (1) Annotation: as each phase exhibits distinct contrast enhancement (representing each phase as a separate domain), the manual annotation (i.e., ground truth image) of all phase images in multi-phase CT becomes a resource-intensive and time-consuming task for liver or tumor segmentation [9]. (2) Poor contrast: some phase images may have poor contrast [10], making it difficult to distinguish the liver boundary. In this paper, we propose a boundary-enhanced liver segmentation network for multi-phase CT images integrated with unsupervised domain adaptation (UDA). The first significant contribution is that we propose a dual discriminator-based unsupervised domain adaptation (DD-UDA) utilizing adversarial learning for liver segmentation on multi-phase images without multi-phase annotation to address the annotation problem. UDA [11] techniques have successfully tackled domain shift problems [12] and have proven to be an effective approach. In [13], an adversarial-based UDA framework comprising a task-specific generator and a discriminator was introduced to obtain an output of the target images that was more similar to the source image, leading to reduced annotation requirements. This approach serves as our baseline method in this paper. While understanding the effectiveness of the baseline approach, it is important to account for the importance of the generator and the discriminator as well.

In order to effectively overcome the domain shift [14,15] challenges encountered in multi-phase CT imaging and to enhance segmentation accuracy, we introduce a pioneering framework called dual discriminator-based unsupervised domain adaptation (DD-UDA). This framework leverages adversarial learning techniques for liver segmentation on multi-phase CT images. To enhance accuracy, we adopt a two-level domain adaptation approach with two discriminators, one focusing on the feature level and the other on the output level. This framework is designed to reduce the discrepancy in distributions between the source and target domains. The second contribution is that we introduce an additional boundary-enhanced decoder network to the encoder–decoder backbone segmentation network (ex., U-Net [5]) to effectively recognize the boundary region, thereby solving the poor contrast problem and improving accuracy. Our private MPCT-FLL CT image dataset comprises various phases, namely, NC, ART, and PV. However, the NC phase image exhibits poor/low contrast [2]), making it extremely challenging to distinguish between different organs. Additionally, certain images in the dataset may suffer from blurring due to issues during the acquisition process. Moreover, the issue of contrast persists when utilizing a multi-center dataset. Consequently, addressing the problem of poor contrast can become highly demanding. In our proposed method, we introduce an additional decoder that extracts the boundary map. By training the network with boundary maps in addition to the feature map and heat maps, we aim to enhance the accuracy when identifying liver regions. The proposed segmentation network has one encoder and two decoders. The design used for the encoder and decoders takes the form of a U-Net-like structure with skip connections, as shown in Figure 2. The encoder and decoder-1 are used to perform segmentation by producing the a heat map for the input image and calculating the supervised segmentation loss. Then, the output of the encoder is fed to decoder-2 and a Sobel filter [16] is applied to retrieve the boundary map. After the output is obtained, the boundary loss for the input image can be calculated. We conducted several ablation experiments using different networks and domain adaptive losses. To conduct the experiments in a domain-adaptive setting, we used the public LiTS dataset [17] as the source domain with annotation, which has only PV phase images, while for the PV, ART, and NC phases we separately used our private MPCT-FLL dataset as the target domain without annotation. Our experimental results show that the proposed framework consisting of a boundary-enhanced segmentation network and dual discriminators achieves improved results comparable with the other state-of-the-art methods. The major contributions of this work are as follows:We propose a novel domain adaptation framework [18] with dual discriminators operating at two levels, incorporating adaptation both at the feature level and at the output level.We introduce an additional boundary-enhanced decoder to determine the boundary loss along with the segmentation loss. The advantage of this approach is that during inference this additional decoder can be dropped, and use the encoder and decoder-1 network to perform liver segmentation. This approach is cost-effective and can be combined with any network.We propose using a boundary-enhanced segmentation network as the generator of our proposed DD-UDA framework to perform liver segmentation on a well-annotated source domain (public dataset) and an unannotated target domain (private dataset). We performed several experiments, and our experimental results show that our proposed framework can achieve significantly improved results compared to other state-of-the-art methods.

Preliminary work was presented as a four-page conference paper at the 44th Annual International Conference of the IEEE: Engineering in Medicine and Biology Society (EMBC 2022) [18]. The paper involved methodological and experimental extensions and validations. As an extension of this conference paper [18], in the present paper we introduce an additional boundary-enhanced decoder to the encoder–decoder backbone segmentation network (ex., U-Net) to further improve its accuracy and solve the poor contrast problem by enhancing boundary regions.

## 2. Related Works

### 2.1. Deep Learning-Based Segmentation Method

The usefulness of deep learning in computer vision has seen significant increases. For detection and segmentation tasks, deep convolutional neural networks (CNNs) have displayed remarkable performance in numerous clinical applications [2]. A fully convolutional neural network (FCN) was introduced in [6] to carry out semantic segmentation. In contrast to FCN, the encoder–decoder-based U-Net network proposed by Ronneberger et al. [5] uses skip connections during upsampling. U-Net was specially developed to conduct segmentation on biomedical images in cases with a dearth of training images. Later, the nested U-Net design known as U-Net++ was proposed by Zhou et al. [19] to minimize the semantic gap between the feature maps produced by the encoder and decoder. In U-Net++, each node is linked to every other node. In order to learn the low-level features, Huang et al. [8] developed U-Net3+, an upgraded version of U-Net++, using full-scale skip connections and deep supervision. While the number of network parameters is decreased by half when utilizing U-Net3+, it is able to achieve increased accuracy. The LiTS liver dataset [17] has been used in research to demonstrate the efficiency of U-Net3+. Utilizing single-phase CT images [6,7] as well as multi-phase CT images [20,21,22], a number of deep learning-based applications have been developed for the segmentation of liver images. These deep learning techniques have two main disadvantages: (1) they require a massive amount of training data as well as the ground truth images in order to train the model; and (2) their accuracy is drastically decreased when the trained model is tested on other datasets due to domain shift problems. Computer vision researchers have developed several additional techniques to address the aforementioned problems.

### 2.2. Unsupervised Domain Adaptation

Domain adaptation is a potential solution to address the above-mentioned problems; it is a part of transfer learning [23], an active subject of research that involves using the source domain as the target domain. There are three types of domain adaptation: supervised, semi-supervised, and unsupervised [12]. Recent studies have shown that UDA is a practical approach for dealing with a scarcity of annotated data in the target domain [24]. The maximum mean discrepancy (MMD) distance measure was proposed in [15] to bridge the domain gap between the source and target domain features. Hoffman et al. [14] introduced adversarial-based UDA for semantic segmentation; the drawback of this approach is that it can only be applied to minor domain shift problems. In order to generate feature maps for the target domain close to the source domain, Tzeng et al. [25] employed a domain discriminator to predict labels generated from the respective domains. Their method exhibited enhanced performance in scenarios where a substantial domain shift is present. Various other methods have been presented for different medical applications: in segmentation-based mammography detection, domain adaptation was performed using a generator and a single discriminator [13]; in fundus image segmentation, domain adaptation was perfomed using the gradient reversal layer (GRL) [26]; and in knee images, segmentation was performed using the UDA approach with a mixup strategy [27]. These methods have all been developed based on using the multilevel UDA approach to adapt and learn the features of semantic segmentation. The existing UDA methods perform adaptation at output level, which can address both the manual annotation and domain shift problems. However, using existing approaches such as [28] with multi-phase CT images achieves poor results in all phases. In [29], unsupervised domain adaptation was performed using the maximum square loss; however, it is important to note that such methods may not be equally effective for all phases of multi-phase CT images due to contrast differences. The strengths and the limitations of the existing methods and our proposals set out in this paper are illustrated in Table 1.

## 3. Method

In this section, we discuss the formal description and the network architecture of our proposed method, as shown in Figure 2.

**Figure 2 bioengineering-10-00899-f002:**
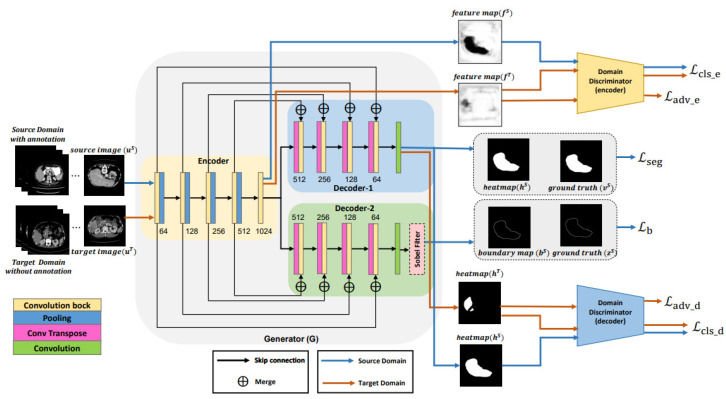
An overview of our proposed boundary-enhanced DD-UDA framework consisting of a boundary-enhanced segmentation network (generator) and dual discriminators.

### 3.1. Formal Description

Assume that *S* is the source domain, which has $N^{S}$ samples, and that each sample consists of an input image $u^{S}\in I_{S}$ along with its corresponding ground truth $v^{S}\in M_{S}$ and boundary label $z^{S}\in B_{S}$. The following is a representation of the dataset in the source domain *S*: ${\left\{\left(u_{i}^{S},v_{i}^{S},z_{i}^{S}\right)\in I_{S}\times M_{S}\times B_{S}\right\}}_{i=1,2,\ldots ,N^{S}}$, where $u_{i}^{S}$ is the *i*th input image in the input space $I_{S}=R^{W\times H\times C}$ (in this paper, the number of channels is C = 1), each pixel value of the ground truth $v_{i}^{S}\;in\;M_{S}=R^{W\times H}$ indicates whether the corresponding pixel of image $u_{i}^{S}$ is liver or not, and the corresponding boundary label $z^{S}\;in\;B_{S}=R^{W\times H}$ is generated by applying the Sobel filter to the ground truth image $v_{i}^{S}$.

On the other hand, the target domain *T* has $N^{T}$ samples, including only input images $u^{T}$ in $I_{T}=R^{W\times H\times C}$. The dataset in the target domain *T* is represented as ${\left\{u_{i}^{T}\in I_{T}\right\}}_{i=1,2,\ldots ,N^{T}}$.

We propose a DD-UDA framework consisting of a segmentation network serving as the generator *G* along with two discriminators $D_{e}$ and $D_{d}$. Additionally, we introduce an additional boundary-enhanced decoder network to the encoder–decoder backbone segmentation network. *G* incorporates a single encoder and two decoders. Multi-task learning can be used to perform multiple tasks simultaneously, improving the generalization capability of the network [30].

The output of the encoder produces feature maps for the source and target domains, which are denoted as $f^{S}=G_{\mathrm{enc}}\left(u^{S}\right)$ and $f^{T}=G_{\mathrm{enc}}\left(u^{T}\right)$, respectively. Decoder-1 produces heat maps for the source and target domains, which are expressed as $h^{S}=G_{\mathrm{dec}1}\left(G_{\mathrm{enc}}\left(u^{S}\right)\right)$ and $h^{T}=G_{\mathrm{dec}1}\left(G_{\mathrm{enc}}\left(u^{T}\right)\right).$ First, we compute the supervised segmentation loss $L_{\mathrm{seg}}$ for the well-annotated source domain *S* (see Section 3.2.1). The boundary maps for the source domain are produced by applying a Sobel filter to the output of decoder-2, represented as follows: $b^{S}=\mathrm{Sobel}\left(G_{\mathrm{dec}}2\left(G_{\mathrm{enc}}\left(u^{S}\right)\right)\right)$, where Sobel(·) is the Sobel filter [16]. The boundary loss $L_{b}$ is only computed for the source domain *S* (see Section 3.2.2). We introduce two discriminators $D_{e}$ and $D_{d}$ to minimize the domain gap between the source and the target domain (see Section 3.3). To bring the output of the target domain much closer to that of the source domain, we compute the adversarial loss for the target domain $L_{\mathrm{adv}\_e}$ and $L_{\mathrm{adv}\_d}$ at the feature and output levels, respectively (see Section 3.3.1). In order to distinguish the outputs of the source and target domain, we calculate the classification loss $L_{\mathrm{cls}\_e}$ and $L_{\mathrm{cls}\_d}$ at the feature and output levels, respectively (see Section 3.3.2).

### 3.2. Boundary-Enhanced Segmentation Network (Generator)

We propose a pixel-to-pixel fully convolutional boundary-enhanced segmentation network consisting of a single encoder and two decoders to achieve an accurate pixel-wise output. The encoder and decoder-1 generate a heat map with the same size as the input image, then the same encoder generates the boundary map using decoder-2.

*Encoder*: The encoder network is comprised of five blocks, each containing two 3 × 3 convolution layers followed by a batch normalization (BN), as well as a ReLU layer in the case of 2D image segmentation. To decrease dimensionality, a 2 × 2 max-pooling operation is introduced after the double convolutional layer in all blocks except the last encoder layer. Furthermore, after each downsampling step the feature channels are doubled.

*Decoder-1 and Decoder-2*: In our proposed approach, we use two separate decoders with the same structure; one is used to perform segmentation and the other for boundary extraction. In the latter, upsampling is performed followed by a 2 × 2 transpose convolutional layer with a stride of 2, reducing the number of feature channels by half. Then, a merge operation is carried out using the previous results from the corresponding encoder block and two 3 × 3 convolutional layers are added, followed by a ReLU and batch normalization layer.

*Classification Layer*: 1 × 1 convolution is used at the last layer of both decoder-1 and decoder-2 to map the channels to the desired number of classes. A Sobel filter [16] is applied to the output of decoder-2 to generate the boundary maps. We use sigmoid as the activation function. Table 2 illustrates the architecture of the proposed boundary-enhanced segmentation network (generator).

To effectively learn and transfer knowledge from the source domain, both the supervised segmentation loss $L_{\mathrm{seg}}$ and the boundary loss $L_{b}$ are calculated.

#### 3.2.1. Segmentation Loss

To learn the features from the well-annotated source domain, the segmentation loss $L_{\mathrm{seg}}$ is calculated on the heat maps $h^{S}=G_{\mathrm{dec}1}\left(G_{\mathrm{enc}}\left(u^{S}\right)\right)$ generated for the source domain using the encoder and decoder-1. The following equation depicts the segmentation loss $L_{\mathrm{seg}}$:(1)$$L_{\mathrm{seg}}\left(u^{S}\right)=\frac{1}{N}\sum_{j=1}^{N}l(v^{S}\left(j\right),h^{S}\left(j\right)).$$
where $v^{S}$ is the corresponding ground truth image, $h^{S}\left(j\right)\in \left[0,1\right]$, *j* is the index of the pixel, *l* indicates the loss function per pixel, and *N* denotes the number of pixels in the image *u*. To handle the data imbalance problem that emerges during the conversion of CT image slices into 2D images, the dice loss (DL) is employed:(2)$$\mathrm{DL}=1-\frac{2|v^{S}\;\cap \;h^{S}|}{\;(\left|v^{S}\right|\;+\;|h^{S}|)\;}.$$

After the segmentation loss $L_{\mathrm{seg}}$ has been calculated, the gradients are backpropagated to the encoder and decoder-1 of the generator.

#### 3.2.2. Boundary Loss

The boundary loss $L_{b}$ is calculated using the boundary maps (i.e., $b^{S}=\mathrm{Sobel}\left(G_{\mathrm{dec}2}\left(u^{S}\right)\right)$) generated by applying the Sobel filter [16] to the output of decoder-2 for the source domain. The boundary loss can be represented as
(3)$$L_{b}\left(u^{S}\right)=\frac{1}{N}\sum_{j=1}^{N}{\left(z^{S}-b^{S}\right)}^{2}.$$

In Equation (Equation 3), $z^{S}$ denotes the corresponding boundary labels. After calculating the boundary loss, the gradients are backpropagated to the encoder and decoder-2 of the generator.

The output generated by the image of the target domain differs from the output generated by the source domain. To address this, we perform UDA using dual discriminators. Adversarial loss and classification loss are employed at two levels to ensure that the output of the target domain resembles that of the source domain.

### 3.3. Dual Discriminator-Based Unsupervised Domain Adaptation Using Adversarial Learning

Our proposed framework incorporates two discriminators, $D_{e}$ and $D_{d}$, to align the feature maps and heat maps, respectively. Both discriminators follow an identical architecture, as shown in Table 3, employing a fully convolutional network. The fully convolutional network offers the flexibility to handle feature map heatmaps of varying sizes. It consists of five layers with a 4 × 4 kernel and a stride of 2, where the number of channels in each layer follows the sequence {64, 128, 256, 512, 1}. Each convolutional layer except the first and last layers is followed by a batch normalization layer. Furthermore, a leaky ReLU layer with a value of 0.2 follows every convolutional layer except for the final one, as discussed in [31].

To reduce the disparity in domain output distribution between the source and target domains, we leverage the encoder and decoder-1 outputs of the generator to compute both the adversarial loss and classification loss. To calculate these losses we assign specific values to each domain, such as α=1 for the source domain and α=0 for the target domain.

#### 3.3.1. Adversarial Loss

The primary objective of employing adversarial-based training is to generate a target domain output (*T*) that closely resembles the source domain (*S*) [13]. The adversarial loss is only computed for the target domain. Additionally, to further bridge the domain gap, the feature map which is the output of the encoder and the heat map which is the output of decoder-1, both generated for the target image, are used as the inputs for the domain discriminators $D_{e}$ and $D_{d}$, respectively. The adversarial loss is then calculated by interchanging the domain values (α=0 for the source domain and α=1 for the target domain). The domain values are interchanged with the aim of confusing the discriminator.

When using feature maps, the adversarial loss $L_{\mathrm{adv}\_e}$ is formulated as follows:(4)$$\begin{array}{c} L_{\mathrm{adv}\_e}\left(F^{T}\right)=\frac{1}{N^{de}}\sum_{j=1}^{N^{de}}l(\alpha ,D_{e}\left(f^{T}\right))\\ =-\frac{1}{N^{de}}\sum_{j=1}^{N^{de}}\log(D_{e}\left(f^{T}\right)). \end{array}$$

When using heatmaps, the adversarial loss $L_{\mathrm{adv}\_d}$ is formulated as follows:(5)$$\begin{array}{c} L_{\mathrm{adv}\_d}\left(H^{T}\right)=\frac{1}{N^{dd}}\sum_{j=1}^{N^{dd}}l(\alpha ,D_{d}\left(h^{T}\right))\\ =-\frac{1}{N^{dd}}\sum_{j=1}^{N^{dd}}\log(D_{d}\left(h^{T}\right)). \end{array}$$

In Equations (Equation 4) and (Equation 5), $f^{T},h^{T}\in \left[0,1\right]$ and $N^{de}$ and $N^{dd}$ indicate the number of pixels in the feature map and heat map of the domain discriminators $D_{e}$ and $D_{d}$, respectively; moreover, *l* denotes the loss function per pixel and adopts the cross entropy loss. After computing $L_{\mathrm{adv}\_e}$ and $L_{\mathrm{adv}\_d}$ the loss is updated to the encoder and decoder-1.

#### 3.3.2. Classification Loss

The classification loss is calculated to differentiate the output of the source domain and the target domain [25]. We employ two discriminators, $D_{e}$ and $D_{d}$, to enhance the classification performance. The classification loss is carefully calculated in two distinct stages to ensure accurate results without interchanging the domain values, i.e., we assign α=1 to the source domain and α=0 to the target domain during computation of the classification loss. Initially, the classification loss $L_{\mathrm{cls}\_e}$ is utilized to effectively differentiate the encoder’s output, specifically, the feature maps $f^{S}$ and $f^{T}$ corresponding to the source domain *S* and the target domain *T*, respectively. Subsequently, the discriminator $D_{e}$ is updated based on the computed classification loss $L_{\mathrm{cls}\_e}$. We represent the classification loss $L_{\mathrm{cls}\_e}$ as shown below:(6)$$\begin{array}{c} L_{\mathrm{cls}\_e}\left(F^{S},F^{T}\right)=L_{\mathrm{cls}\_e}\left(F^{S}\right)+L_{\mathrm{cls}\_e}\left(F^{T}\right)\\ =-\frac{1}{N^{de\_S}}\sum_{j=1}^{N^{de\_S}}\alpha \;\log\left(D_{e}\left(f^{S}\right)\right)\\ -\frac{1}{N^{de\_T}}\sum_{j=1}^{N^{de\_T}}\left(1-\alpha \right)\log\left(D_{e}\left(f^{T}\right)\right).\; \end{array}$$

Similarly, we utilize the classification loss $L_{\mathrm{cls}\_d}$ to distinguish the output of decoder-1, specifically, the heatmaps $h^{S}$ and $h^{T}$ generated for the source domain *S* and the target domain *T*, respectively. After the computation of the classification loss $L_{\mathrm{cls}\_d}$ is complete, the discriminator $D_{d}$ is updated accordingly. The classification loss $L_{\mathrm{cls}\_d}$ can be expressed as follows:(7)$$\begin{array}{c} L_{\mathrm{cls}\_d}\left(H^{S},H^{T}\right)=L_{\mathrm{cls}\_d}\left(H^{S}\right)+L_{\mathrm{cls}\_d}\left(H^{T}\right)\\ =-\frac{1}{N^{dd\_S}}\sum_{j=1}^{N^{dd\_S}}\alpha \;\log\left(D_{d}\left(h^{S}\right)\right)\\ -\frac{1}{N^{dd\_T}}\sum_{j=1}^{N^{dd\_T}}\left(1-\alpha \right)\log\left(D_{d}\left(h^{T}\right)\right).\; \end{array}$$

### 3.4. Objective Function

The structure of our framework is visualized in Figure 2. Initially, during training, we employ the heat map, which is the output of decoder-1, to calculate the segmentation loss for the source domain. In addition, we utilize the boundary map, which corresponds to the output of decoder-2, to compute the boundary loss. Equations (Equation 1) and (Equation 3) provide a detailed representation of these calculations. The adversarial loss is determined for the target domain; this computation occurs at both the feature level and the output level, as described in Equations (Equation 4) and (Equation 5). The classification loss is computed by utilizing the source and target domains at the feature and output levels, as described in Equations (Equation 6) and (Equation 7). The objective function can be written as shown below:(8)$$L_{\mathrm{obj}}=L_{\mathrm{seg}}+L_{b}\times \lambda_{b}+L_{\mathrm{adv}\_e}\times \lambda_{\mathrm{adv}}\_e+L_{\mathrm{adv}\_d}\times \lambda_{\mathrm{adv}}\_d+L_{\mathrm{cls}\_e}+L_{\mathrm{cls}\_d}.\;$$

In Equation (Equation 8), we represent the weight as λ, where $\lambda_{b}$ is fixed at 0.2; $\lambda_{b}$ is chosen as 0.2 in order to normalize the segmentation result. We attempted the calculation using 0.15; however, we tended to achieve better results using 0.2. To ensure similarity between the outputs of the target domain and the source domain, we employed the respective adversarial weights $\lambda_{\mathrm{adv}}\_e$ = 0.0005 and $\lambda_{\mathrm{adv}}\_d$ = 0.003 for computation of the adversarial loss. The adversarial weight $\lambda_{\mathrm{adv}}\_e$ = 0.0005 was selected based on the baseline method [13] and experiments conducted through trail and error, as shown in the table below. We initially tried to perform the experiment for the DD-UDA approach by fixing the adversarial weights $\lambda_{\mathrm{adv}}\_e$ and $\lambda_{\mathrm{adv}}\_d$ to 0.0005. However, using these values led to improved results only for the ART and NC phases, not for the PV phase. Thus, we fixed $\lambda_{\mathrm{adv}}\_e$ to 0.0005 and carried out different experiments with different $\lambda_{\mathrm{adv}}\_d$, such as 0.003 and 0.007. From the numerical results illustrated in Table 6, it can be observed that the best results were achieved with $\lambda_{\mathrm{adv}}\_d$ set to 0.003 and $\lambda_{\mathrm{adv}}\_e$ set to 0.0005.

## 4. Results

### 4.1. Dataset

For our experimental setup, the Liver Tumor Segmentation (LiTS) dataset [17], which is publicly available, was utilized as the source domain *S*. This dataset consists of PV phase images along with corresponding ground truth images of 131 patients. The number of slices ranges from 46 to 1026. To train our framework, we used images and corresponding ground truth for 111 of 131 patients.

We employed each phase (PV, ART, and NC) from our private Multi-Phase CT–Focal Liver Lesions (MPCT-FLL) dataset as the target domain. This dataset was provided by Sir Run Run Shaw Hospital, Zhejiang University, Hangzhou, China. This dataset consists of the data of 121 patients. The number of slices ranges from 20 to 99. To train our framework, we utilized the data for 110 of 121 patients, including PV, ART, and NC phase images of the target domain without ground truth images. To evaluate the performance of our proposed method, we used the test data of the remaining eleven patients along with the ground truth images. The size of the CT images was 512 × 512 in both datasets. Table 4 and Table 5 show the details of the two datasets.

### 4.2. Data Preprocessing

Typically, CT images are 3D images; in our research, however, we utilize each slice of the CT image as a 2D image. To enhance the CT image, we use a windowing operation. Enhanced brightness and contrast can be achieved by adjusting the window length (WL) and window width (WW). We set WL = −20 and WW = 200. While pixels greater than 200 are set to 200, those less than −20 were set to −20. We used min-max normalization to rescale the features between 0 and 1. Figure 3 illustrates the data preprocessing operation.

### 4.3. Implementation Details

Our experiments were performed using a GPU RTX 8000 equipped with 48GB of memory. The proposed framework was developed using PyTorch, and backpropagation was carried out using the Adam optimizer [32]. The generator’s learning rate was set to 1 × 10${}^{-5}$, while the two discriminators were fixed to 1 × 10${}^{-6}$ and the batch size was set to 8. We trained our proposed framework for 50 epochs when using the public dataset [17] as the source and the private dataset as target. When performing the experiments using the PV phase of our private dataset as the source and other phases of private dataset as the target, we trained our framework for 200 epochs. For testing, we used only the encoder and decoder-1 to perform segmentation, eliminating the additional decoder-2. Hence, this approach is more cost-effective and can be combined with other encoder–decoder-based networks.

### 4.4. Training Strategy

In this paper, we propose a boundary-enhanced UDA framework with dual discriminators. During each iteration of training, we provided the framework with a source image and its corresponding ground truth, boundary labels, and target image, denoted as $\left(u_{i}^{S},v_{i}^{S},z_{i}^{S},u_{i}^{T}\right)$. In each iteration, five major steps are involved in training our proposed framework: (1) Utilizing the heat map generated as the output of decoder-1 for the source image, the supervised segmentation loss $L_{\mathrm{seg}}$ is computed. After computation, the encoder and decoder-1 are updated. (2) The boundary loss $L_{b}$ generated for the source image is calculated by applying a Sobel filter to the output of decoder-2. The $L_{b}$ is then backpropagated for the encoder and decoder-2. (3) Because we do not have access to the corresponding ground truth images, the feature map and heat maps generated by the encoder and decoder-1 of the generator are employed for the target domain *T*. These outputs are then fed as inputs to the discriminator $D_{e}$ and $D_{d}$, respectively. The respective adversarial losses $L_{\mathrm{adv}}\_e$ and $L_{\mathrm{adv}\_d}$ are computed by incorporating adversarial weights and exchanging the values from domain [18] for those of their respective domains. To ensure that the target domain output matches that of the source domain, the losses are backpropagated to update the encoder and decoder-1. (4) The feature maps for the source and target domains are provided as input to the discriminator $D_{e}$. The classification loss $L_{\mathrm{cls}\_e}$ is computed by assigning the respective domain values, and updating is performed only for $D_{e}$. (5) During the final step, the heat maps for the source and target domains are provided as input to the discriminator $D_{d}$ with their respective domain values assigned. The classification loss $L_{\mathrm{cls}\_d}$ is computed and the discriminator $D_{d}$ is updated accordingly. We utilize steps (4) and (5) to differentiate the feature maps and heat maps of the source and target domains in their respective domains. These steps help to identify and separate the unique characteristics and spatial information of each domain’s feature maps and heat maps. In this way, it is possible to better understand and analyze the differences between the source and target domains, enabling effective domain-specific learning and adaptation.

### 4.5. Evaluation

We assess the effectiveness of our proposed method by employing three evaluation metrics: the dice coefficient (DC), intersection over union (IoU), sensitivity (TRP), and precision (PPV).

#### 4.5.1. Dice Coefficient (DC)/F1 Score

The dice coefficient (DC) is a metric used to measure the similarity between the ground truth *v* and the predicted output *h*. It quantifies the extent of overlap between these two by calculating twice the area of overlap divided by the total number of pixels in both *v* and *h*. Higher values of the dice coefficient indicate better segmentation results. The DC is calculated using the formula shown below:(9)$$\mathrm{DC}=\frac{2|v\cap h|}{\left|v|+|h\right|}.$$

#### 4.5.2. Intersection over Union (IoU)/Jaccard

The intersection over union (IoU) is a metric that measures the overlap between the ground truth *v* and the predicted output *h*. It is computed by dividing the area of overlap by the area of union between *v* and *h*. The IoU value ranges from 0–1. If the accuracy is closer to 1, this indicates greater similarity between the predicted output and the ground truth. The formula used to calculate IoU is
(10)$$\mathrm{IoU}=\frac{\left|v\cap h\right|}{\left|v\cup h\right|}.$$

#### 4.5.3. Sensitivity/True Positive Rate (TRP)/Recall

The sensitivity measures the proportion of positives that are correctly segmented, which can be defined as follows:(11)$$\mathrm{TRP}=\frac{\mathrm{TP}}{\mathrm{TP}+\mathrm{FN}}.$$
where TP (true positive) is the number of liver pixels correctly predicted to be liver and FN (false negative) is the number of liver pixels identified as background.

#### 4.5.4. Precision/Positive Predicted Value (PPV)

The precision score is the number of true positive results divided by the number of all positive results. The formula used to calculate PPV is
(12)$$\mathrm{PPV}=\frac{\mathrm{TP}}{\mathrm{TP}+\mathrm{FP}}.$$
where TP (true positive) is the number of liver pixels correctly predicted to be liver and FP (false positive) is the number of background pixels wrongly recognized as liver.

### 4.6. Performance Evaluation of the Training Model

This section provides a detailed description of the training process of our proposed network. The network utilizes well-annotated private PV data as the source domain, while our unannotated private ART data serve as the target domain for training. To illustrate this process, Figure 4a showcases a sample of training input images and Figure 4b visually demonstrates the convergence achieved during the training phase, showcasing the network’s ability to learn. From Figure 4b, it can be observed that the model is trained smoothly and converges at 200 epochs during the training. Finally, Figure 4c displays the predicted output generated by the network for the training images, emphasizing the network’s capacity to produce accurate results based on the training data. A DC score of 0.981 was achieved while training the network.

### 4.7. Evaluation of Proposed DD-UDA Framework Based on Hyperparameters Such as Adversarial Weights

In this experiment, we used only well-annotated public PV data as the source and unannotated private PV data as the target domain for training. We tested the model on our private PV dataset. A domain gap [25] exists in this case because both datasets are from different data centers, even though they have the same phase. In a similar experiment, we used only well-annotated public PV data as the source and our unannotated private ART data as the target domain for training. We tested the model on our private ART dataset. In this case, a domain gap exists because the datasets are from different data centers and have different phases. In another variation, we used only well-annotated public PV data as the source and our unannotated private NC data as the target domain for training. We tested the model on our private NC dataset. The domain gap exists because both datasets are from different data centers have different phases.

To assess the effectiveness of our proposed methods based on adversarial weights [18], we conducted the experiments using different combinations of adversarial weights, as shown in Table 6. From Table 6, it can be observed that improved results were achieved for the PV, ART and NC phases when $\lambda_{\mathrm{adv\_e}}$ was 0.0005 and $\lambda_{\mathrm{adv\_d}}$ was 0.003.

**Table 6 bioengineering-10-00899-t006:** Evaluation of proposed DD-UDA framework based on hyperparameters such as adversarial weights.

Proposed DD-UDA	$\lambda_{\mathrm{adv\_e}}$	$\lambda_{\mathrm{adv\_d}}$	PV(LiTS)→ PV(MPCT-FLL)	PV(LiTS)→ ART(MPCT-FLL)	PV(LiTS)→ NC(MPCT-FLL)
			**DC**	**DC**	**DC**
	0.0005	0.0005	0.885	0.887	0.855
	0.0005	0.003	0.894	0.888	0.872
	0.0005	0.007	0.872	0.866	0.845

### 4.8. Ablation Study

To assess the efficacy of our proposed unsupervised domain adaptation approaches, we conducted the following experiments.

#### 4.8.1. Different Datacenter and Same Phase

We used only well-annotated public PV data [17] as the source domain and unannotated private PV data as the target domain for training. We tested the model on our private PV dataset. A domain gap exists in this experiment; although they have the same phase, the datasets are from different data centers. The outcome of this ablation study is illustrated in Table 7, while Figure 5 shows how accurately the liver region can be segmented using the experiments mentioned above.

#### 4.8.2. Same Datacenter and Different Phase

In this experiment, we used only well-annotated private PV data as the source domain, and used our unannotated private ART data as the target domain for training. We tested the model on our private ART dataset. A domain gap [25] exists in this case because, although both datasets are from the same data center, they have different phases. In a similar experiment, we used only well-annotated private PV data as the source domain and used our unannotated private NC data as the target domain for training. We tested the model on our private NC dataset. Again, a domain gap exists because while both datasets are from the same data center, they have different phases. Table 8 demonstrates the results of this ablation study.

#### 4.8.3. Different Datacenter and Different Phase

In this experiment, we used only well-annotated public PV data [17] as the source domain, and used our unannotated private ART data as the target domain for training. We tested the model on our private ART dataset. A domain gap exists because the datasets are from different data centers and different phases. In a similar experiment, we used only well-annotated public PV data as the source domain and used our unannotated private NC data as the target domain for training. We tested the model on our private NC dataset. Again, a domain gap exists because both datasets are from different data centers and different phases. Table 9 demonstrates the experimental outcomes of this ablation study.

Initially, U-Net [5] was used to perform segmentation using well-annotated public PV phase data, achieving an IoU of 0.728 and 0.736 for the private ART and public NC phase data, respectively. Our proposed boundary-enhanced U-Net with two decoders to perform segmentation without domain adaptation achieved an improved IoU of 0.753 and 0.749. Despite achieving better results than the conventional U-Net, our proposed boundary-enhanced U-Net has a limitation in that it relies on a supervised learning strategy, demanding a large amount of annotated training data during the training process. Moreover, when compared to the UDA approach this method does not perform as effectively on unknown target domains.

We then employed the U-Net+UDA [13] approach, which is the baseline method considered in this paper, achieving IoUs of 0.796 and 0.772. For our proposed boundary-enhanced U-Net with the UDA approach, the IoU values improve to 0.789 and 0.781. The IoU values achieved by our proposed U-Net+DD-UDA approach [18], which performs UDA using two discriminators, are 0.808 and 0.794. These values are further improved to 0.811 and 0.800 when using our proposed boundary-enhanced U-Net as a generator for the proposed DD-UDA framework. Figure 6 and Figure 7 show the segmentation accuracy for liver region achieved in the experiments mentioned above.

### 4.9. Evaluation of Proposed Methods with SegNet as the Backbone

To assess the effectiveness of our proposed methods, we conducted an experiment utilizing SegNet [33] as the backbone network. First, we used only well-annotated public PV data as the source and unannotated private PV data as the target domain for training. We tested the model on our private PV dataset. A domain gap [25] exists because both datasets are from different data centers, even though they have the same phase. In a similar experiment, we used only well-annotated public PV data as the source and our unannotated private ART data as the target domain for training. We tested the model on our private ART dataset. In this case, a domain gap exists because the datasets are from different data centers and have different phases. Next, we used only well-annotated public PV data [17] as the source and our unannotated private NC data as the target domain for training. We tested the model on our private NC dataset. Again, a domain gap exists because the datasets are from different data centers and have different phases. Table 10 demonstrates the numerical results for our proposed methods. From Table 10, it can be seen that our proposed methods show improved results when using SegNet as the backbone.

### 4.10. Comparison with the State-of-the-Art Methods

Finally, we compared our results with other state-of-the-art methods. The numerical outcomes are presented in Table 11, while Figure 8 visually illustrates the segmentation results achieved with our proposed methods in comparison to those achieved with other methods.

*No Adaptation*: First, the outcomes were assessed by performing segmentation without using any domain adaptation. We utilized segmentation networks such as U-Net [5] and U-Net3+ [20] to perform segmentation. PV phase images and corresponding ground truth images from the LiTS dataset were used as training data and tested on each phase of our private dataset. The outcomes obtained with U-Net and U-Net3+ are relatively low for all phases. The tested segmentation networks such as U-Net and U-Net3+ consisted of the encoder–decoder-based segmentation network only.

*Unsupervised Domain Adaptation*: In [13], a UDA approach with adversarial learning was proposed for mass detection in mammograms. In this method, the trainable parameters are the generator, which is a fully convolutional network, and the discriminator. In [28], the authors introduced a weighted adversarial unsupervised domain adaptation by employing a domain discriminator that they called ADVENT. Experiments were performed by converting feature maps into entropy maps. This method uses a segmentation network and a single discriminator, and involves direct entropy minimization. The results obtained with this method are comparatively very low for all the phases of multi-phase CT images. Another approach is to use domain adaptation to address the class imbalance problem using the maximum square loss (MSL) [29], which is the negative sum of the squared probabilities. The MSL is calculated directly from the feature map. This strategy produces improved results for our private PV phase data, while the results achieved for the ART and NC phases are relatively low. In [26], the authors used a GRL layer to perform UDA for biomedical image segmentation using heat maps; this method uses a segmentation network only. This approach is able to achieve better accuracy for the NC phase. In [27], the authors introduced adversarial-based unsupervised domain adaptation with a mixup strategy, which works well for the ART phase. In [34], the authors proposed combining the UDA approach with the MSL to perform segmentation-based detection of multi-phase CT images. They have utilized a detection head and a single discriminator in a method that involves direct entropy minimization. Improved accuracy with this method is seen only in the PV phase. Unlike the above-mentioned approaches, our proposed DD-UDA [18] method is able to achieve significantly improved results for all three phases. The trainable parameters used in our approach consist of an encoder–decoder-based segmentation network and two discriminators. During inference, we make use of the segmentation network only. In this research, U-Net was employed as the backbone in all our comparative experiments with the state-of-the-art methods. From Table 11 and Figure 8, it can be observed that the proposed method achieves significantly more accurate results in comparison to other existing methods. To train of our method, we utilized one encoder and two decoders with two discriminators. Only the segmentation network (i.e., encoder and decoder-1) is employed during inference. This approach can accurately segment the liver region.

## 5. Discussion

The scarcity of annotated data poses a major obstacle when utilizing medical image datasets in the training of deep learning-based methods [5]. Performing segmentation on medical images is considerably more challenging than using natural images, primarily due to the intricate task of differentiating between various organs [1]. Moreover, training deep learning models using multi-phase CT images adds to the complexity and expense, as annotation is required for each image in each different phase. This limitation applies to our proposed boundary-enhanced U-Net, as it requires annotated data to train the network. Furthermore, issues with poor contrast encountered during testing on the ART and NC phases result in significantly diminished performance due to domain shift problems. Although existing UDA methods [13,26] can alleviate the annotation problem, they tend to focus on adaptation solely at the output level. Consequently, UDA approaches such as [28,29] exhibit poor performance, while others demonstrate better results only for specific phases of multi-phase CT images. To address these challenges, in this paper we propose the DD-UDA method, which incorporates an additional boundary-enhanced segmentation network to accurately segment the liver regions.

In our research, we conducted experiments to showcase the efficacy of our proposed methods. Specifically, we evaluated our methods in three different scenarios: (1) using data from different data centers from the same phase; (2) using data from the same data center and different phases; and (3) using data from different data centers and different phases. Table 7, Table 8 and Table 9 illustrate the outcomes of these experiments. From our experimental results, it is evident that the proposed DD-UDA (Domain Adaptation with Dual Discriminators) method outperforms existing UDA (Unsupervised Domain Adaptation) methods [13]. We achieve this improvement by performing domain adaptation at both the feature and output levels utilizing the proposed dual discriminators. Additionally, by incorporating the boundary enhanced decoder, as shown in Figure 5, Figure 6 and Figure 7, the proposed method is able to achieve highly accurate segmented regions that closely resembles the ground truth images. In addition, we explored the impact of replacing the U-Net backbone [5] with the SegNet backbone [33]. Remarkably, even with a different backbone our proposed approach was able to achieve significantly improved performance, highlighting the robustness and versatility of our approach. One of the key strengths of our proposed method lies in its efficient utilization of a private multi-phase dataset without annotated data.

## 6. Conclusions

In this paper, we have proposed a novel DD-UDA framework to improve the existing UDA approach. Domain adaptation is performed at the feature and output levels using a generator and a discriminator. In addition, we propose an additional boundary-enhanced decoder to determine the boundary loss along with the segmentation loss. The advantage of this approach is that during inference this additional decoder can be dropped and the encoder and decoder-1 network can be used to perform liver segmentation. Thus, this approach is cost-effective and can be combined with any network. Our experimental findings illustrate that our proposed method surpasses the existing state-of-the-art approaches in terms of performance. A significant benefit of this approach is its ability to train a deep learning model using multi-phase CT images even in the absence of corresponding ground truth. This advantage significantly alleviates the burden of manual annotation for radiologists. In future work, we intend to develop unsupervised approaches for medical image segmentation by further minimizing the domain gap and increasing the generalization capability.

## Figures and Tables

**Figure 1 bioengineering-10-00899-f001:**
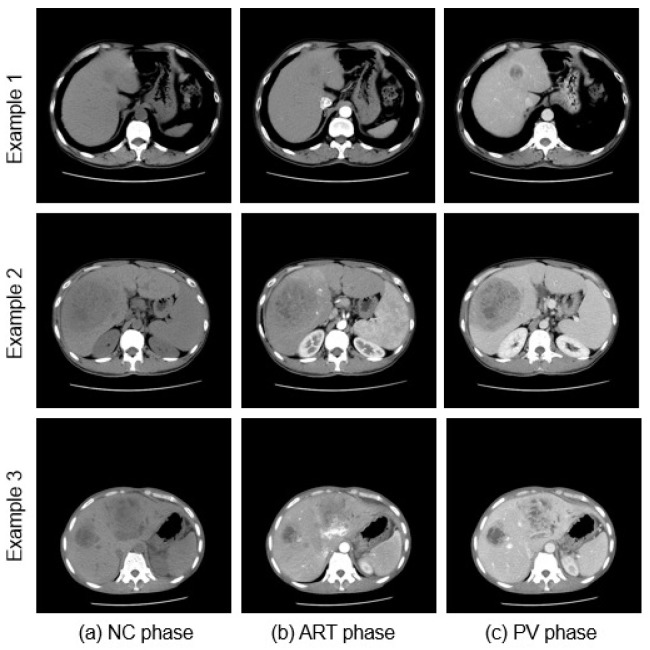
Examples illustrating 2D slices of the NC, ART, and PV phases of corresponding multi-phase CT images taken at different time periods.

**Figure 3 bioengineering-10-00899-f003:**
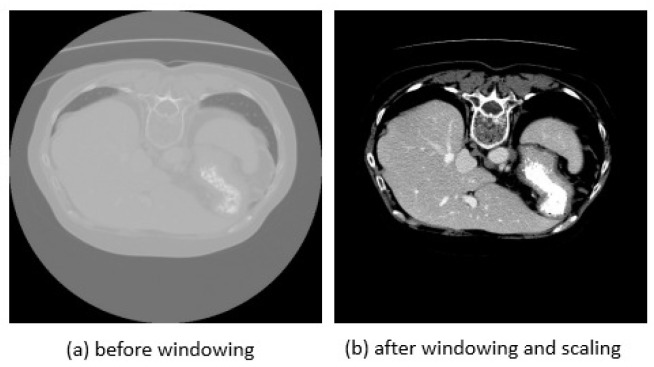
Illustration of data preprocessing operation before and after windowing and scaling operations.

**Figure 4 bioengineering-10-00899-f004:**
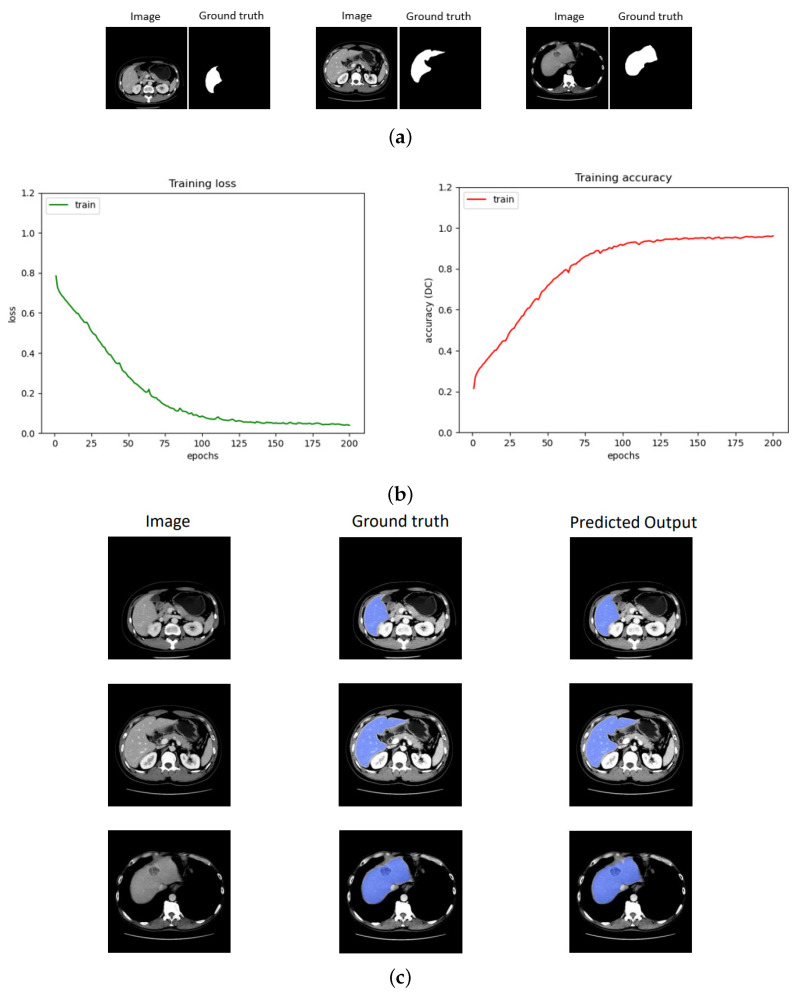
Illustration of the training process and the output prediction: (**a**) sample input images with mask, (**b**) training convergence using loss and accuracy, and (**c**) segmented final results.

**Figure 5 bioengineering-10-00899-f005:**
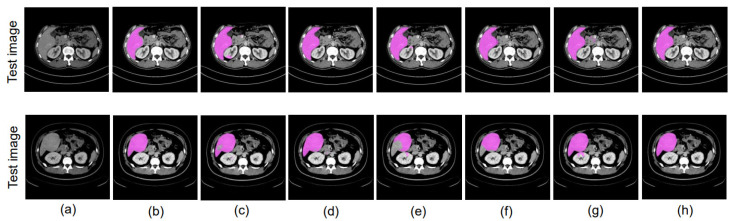
Experimental results using public PV phase as the source domain and private PV phase as the target domain: (**a**) PV phase image, (**b**) ground truth, (**c**) U-Net [5], (**d**) proposed boundary-enhanced U-Net, (**e**) UDA [13], (**f**) boundary-enhanced U-Net+UDA, (**g**) proposed DD-UDA framework, (**h**) proposed boundary-enhanced U-Net+DD-UDA framework.

**Figure 6 bioengineering-10-00899-f006:**
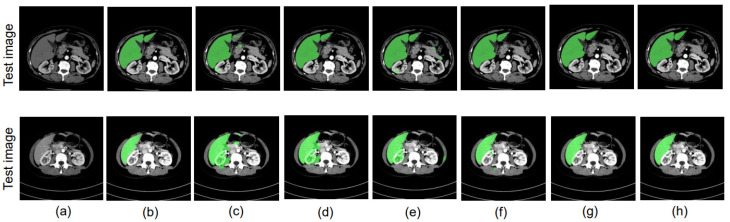
Experimental results using public PV phase as the source domain and private ART phase as the target domain: (**a**) ART phase, (**b**) ground truth, (**c**) U-Net [5], (**d**) proposed boundary-enhanced U-Net, (**e**) UDA [13], (**f**) boundary-enhanced U-Net+UDA, (**g**) proposed DD-UDA framework, (**h**) proposed boundary-enhanced U-Net+DD-UDA framework.

**Figure 7 bioengineering-10-00899-f007:**
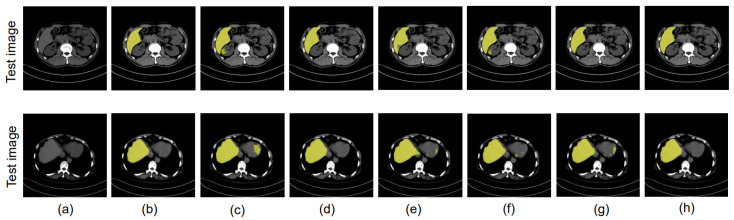
Experimental results using public PV phase as the source domain and private NC phase as the target domain: (**a**) NC phase, (**b**) ground truth, (**c**) U-Net [5], (**d**) proposed boundary-enhanced U-Net, (**e**) UDA [13], (**f**) boundary-enhanced U-Net+UDA, (**g**) proposed DD-UDA framework, (**h**) proposed boundary-enhanced U-Net+DD-UDA framework.

**Figure 8 bioengineering-10-00899-f008:**
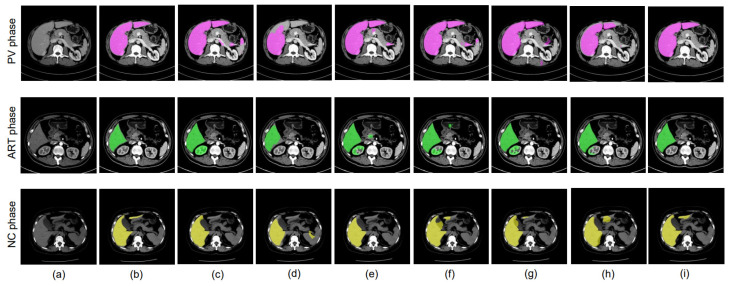
Comparison of segmentation results with other state-of-the-art methods on the PV, ART, and NC phases of our private MPCT-FLL dataset. (**a**) CT image, (**b**) ground truth, (**c**) UDA [13], (**d**) ADVENT [28], (**e**) MSL+IW [29], (**f**) GRL [26], (**g**) MIXUP [27], (**h**) UDA+MSL [34], (**i**) proposed boundary-enhanced U-Net+DD-UDA framework.

**Table 1 bioengineering-10-00899-t001:** Strengths and limitations of the proposed method in comparison with previous works.

Method	Strength	Limitations
U-Net [5]	Can perform segmentation using smaller dataset.	Lower generalization capability when tested on unknown datasets.
UDA [13]	Adaptation is performed at output level using single discriminator.	This method does not show improved results for the PV and NC phases.
UDA(GRL) [26]	Adaptation is performed using GRL layer.	This method achieves better results only for the NC phase.
Advent [28]	Adaptation is performed by converting feature maps to entropy maps.	This approach show poor results on medical datasets.
MSL [29]	Adaptation is performed using the maximum square loss without discriminator.	This approach does not achieve better results for certain phases, and is designed for multi-class segmentation.
Proposed Method	Adaptation is carried out by employing two discriminators, one for each of the feature and output levels. We introduce an additional decoder to effectively learn the boundary features.	Our proposed method requires more trainable parameters.

**Table 2 bioengineering-10-00899-t002:** Architecture of the proposed boundary-enhanced segmentation network (generator).

Encoder		Decoder-1 and Decoder-2	
**Layer**	**Details** **(Kernel Size, Output Channels, BN,** **Leaky Relu, Stride, Padding)**	**Layer**	**Details**
input	CT image	upsample1	2×2 upsample of conv5-2 concatenate with conv4-2
conv1-1	3×3×64, BN, Leaky Relu, 2, 1	conv 6-1	3×3×512 Leaky Relu
conv1-2	3×3×64,BN, Leaky Relu, 2, 1	conv 6-2	3×3×512 Leaky Relu
pool 1	2×2, 2	upsample2	2×2 upsample of conv6-2 concatenate with conv3-2
conv2-1	3×3×128,BN, Leaky Relu, 2, 1	conv 7-1	3×3×256 Leaky Relu
conv2-2	3×3×128,BN, Leaky Relu, 2, 1	conv 7-2	3×3×256 Leaky Relu
pool 2	2×2, 2	upsample3	2×2 upsample of conv7-2 concatenate with conv2-2
conv3-1	3×3×256,BN, Leaky Relu, 2, 1	conv 8-1	3×3×128 Leaky Relu
conv3-2	3×3×256,BN, Leaky Relu, 2, 1	conv 8-2	3×3×128 Leaky Relu
pool 3	2×2, 2	upsample4	2×2 upsample of conv8-2 concatenate with conv1-2
conv4-1	3×3×512,BN, Leaky Relu, 2, 1	conv 9-1	3×3×64 Leaky Relu
conv4-2	3×3×512,BN, Leaky Relu, 2, 1	conv 9-2	3×3×64 Leaky Relu
pool 4	2×2, 2	conv 10	1×1×1
conv5-1	3×3×1024,BN, Leaky Relu, 2, 1		
conv5-2	3×3×1024,BN, Leaky Relu, 2, 1		

**Table 3 bioengineering-10-00899-t003:** Architecture of the discriminator network.

Layers	Output	Operation, Kernel Size Output Channels, Stride
Layer-1	64×64	Conv, 4×4, 64, 2
Layer-2	32×32	Conv, 4×4, 128, 2
Layer-3	16×16	Conv, 4×4, 256, 2
Layer-4	8×8	Conv, 4×4, 512, 2
Layer-5	4×4	Conv, 4×4, 1, 2

**Table 4 bioengineering-10-00899-t004:** Details of our internal MPCT-FLL dataset.

	PV(MPCT-FLL)→ART(MPCT-FLL)			PV(MPCT-FLL)→ NC(MPCT-FLL)		
	Source	Target	Test	Source	Target	Test
No. of Patients	71	39	11	71	39	11
No. of slices	1888	985	257	1888	989	255

**Table 5 bioengineering-10-00899-t005:** Details of the public (LiTS) and private (MPCT-FLL) dataset.

	PV(LiTS)→ PV(MPCT-FLL)			PV(LiTS)→ ART(MPCT-FLL)			PV(LiTS)→ NC(MPCT-FLL)		
	Source	Target	Test	Source	Target	Test	Source	Target	Test
No. of Patients	111	110	11	111	110	11	111	110	11
No. of slices	16,156	2887	274	16,156	2881	257	16,156	2855	255

**Table 7 bioengineering-10-00899-t007:** Ablation study using public PV phase data as the source domain and private PV phase data as the target domain.

Method	Boundary	UDA	DD-UDA	PV(LiTS)→ PV(MPCT-FLL)			
				DC	IoU	TRP	PPV
U-Net [5]				0.866	0.775	0.549	0.788
Proposed Boundary-enhanced U-Net	✓			0.877	0.793	0.546	0.805
U-Net+UDA (baseline) [13]		✓		0.872	0.785	0.916	0.812
Proposed Boundary- enhanced U-Net+UDA	✓	✓		0.884	0.803	0.967	0.823
U-Net+ Proposed DD-UDA [18]			✓	0.894	0.817	0.983	0.828
Proposed Boundary- enhanced U-Net+ DD-UDA	✓		✓	0.892	0.823	0.975	0.830

**Table 8 bioengineering-10-00899-t008:** Ablation study using private PV phase data as the source domain and private ART or NC phase data as the target domain.

Method	Boundary	UDA	DD- UDA	PV(MPCT-FLL)→ ART(MPCT-FLL)				PV(MPCT-FLL)→ NC(MPCT-FLL)			
				DC	IoU	TRP	PPV	DC	IoU	TRP	PPV
U-Net [5]				0.905	0.846	0.543	0.903	0.859	0.785	0.544	0.863
Proposed Boundary -enhanced U-Net	✓			0.913	0.853	0.572	0.908	0.862	0.788	0.584	0.865
U-Net+UDA (baseline) [13]		✓		0.914	0.859	0.878	0.919	0.893	0.829	0.847	0.895
Proposed Boundary- enhanced U-Net+UDA	✓	✓		0.918	0.872	0.844	0.924	0.894	0.852	0.836	0.852
U-Net+ Proposed DD-UDA [18]			✓	0.921	0.863	0.875	0.922	0.911	0.857	0.902	0.903
Proposed Boundary- enhanced U-Net+DD-UDA	✓		✓	0.922	0.883	0.929	0.934	0.912	0.862	0.897	0.924

**Table 9 bioengineering-10-00899-t009:** Ablation study using public PV phase data as the source domain and private ART and NC phase data as the target domain.

Method	Boundary	UDA	DD- UDA	PV(LiTS)→ ART(MPCT-FLL)				PV(LiTS)→ NC(MPCT-FLL)			
				DC	IoU	TRP	PPV	DC	IoU	TRP	PPV
U-Net [5]				0.826	0.728	0.560	0.752	0.825	0.736	0.564	0.741
Proposed Boundary- enhanced U-Net	✓			0.844	0.753	0.537	0.537	0.779	0.749	0.535	0.762
U-Net+UDA (baseline) [13]		✓		0.880	0.796	0.966	0.813	0.855	0.772	0.944	0.811
Proposed Boundary- enhanced U-Net+UDA	✓	✓		0.874	0.789	0.892	0.821	0.864	0.781	0.818	0.811
U-Net+Proposed DD-UDA [18]			✓	0.888	0.808	0.750	0.832	0.872	0.794	0.662	0.821
Proposed Boundary- enhanced U-Net+ DD-UDA	✓		✓	0.890	0.811	0.966	0.838	0.875	0.800	0.945	0.832

**Table 10 bioengineering-10-00899-t010:** Evaluation of proposed methods with SegNet as the backbone.

Method	PV(LiTS)→ PV(MPCT-FLL)				PV(LiTS)→ ART(MPCT-FLL)				PV(LiTS)→ NC(MPCT-FLL)			
	DC	IoU	TRP	PPV	DC	IoU	TRP	PPV	DC	IoU	TRP	PPV
SegNet [33]	0.879	0.796	0.747	0.819	0.868	0.783	0.667	0.809	0.836	0.750	0.632	0.802
Proposed Boundary -enhanced SegNet	0.884	0.805	0.825	0.826	0.861	0.779	0.738	0.822	0.845	0.762	0.681	0.808
SegNet+Proposed DD-UDA [18]	0.890	0.813	0.893	0.830	0.896	0.819	0.966	0.838	0.886	0.811	0.850	0.841
Proposed Boundary- enhanced SegNet+ DD-UDA	0.901	0.828	0.929	0.850	0.899	0.823	0.959	0.843	0.894	0.822	0.933	0.840

**Table 11 bioengineering-10-00899-t011:** Comparison with the state-of-the-art methods.

Method	PV(LiTS)→ PV(MPCT-FLL)				PV(LiTS)→ ART(MPCT-FLL)				PV(LiTS)→ NC(MPCT-FLL)			
	DC	IoU	TRP	PPV	DC	IoU	TRP	PPV	DC	IoU	TRP	PPV
No adaptation												
U-Net [5]	0.866	0.775	0.549	0.788	0.826	0.728	0.560	0.752	0.825	0.736	0.564	0.741
U-Net 3+ [8]	0.878	0.794	0.976	0.809	0.869	0.781	0.960	0.802	0.847	0.758	0.938	0.791
Unsupervised domain adaptation												
UDA [13]	0.872	0.785	0.916	0.812	0.880	0.796	0.966	0.813	0.855	0.772	0.944	0.811
ADVENT [28]	0.706	0.629	0.500	0.706	0.629	0.547	0.500	0.668	0.626	0.538	0.500	0.664
MSL+IW [29]	0.884	0.803	0.978	0.818	0.863	0.777	0.963	0.797	0.833	0.747	0.952	0.813
UDA(GRL) [26]	0.879	0.796	0.981	0.807	0.853	0.769	0.935	0.810	0.863	0.787	0.929	0.826
UDA(MIXUP) [27]	0.870	0.784	0.613	0.812	0.880	0.797	0.960	0.816	0.854	0.766	0.958	0.781
UDA+MSL [34]	0.874	0.790	0.500	0.827	0.863	0.782	0.804	0.808	0.834	0.751	0.909	0.810
Proposed DD-UDA [18]	0.894	0.817	0.983	0.828	0.888	0.808	0.750	0.832	0.872	0.794	0.662	0.821
Proposed Boundary- enhanced U-Net+DD-UDA	0.892	0.823	0.975	0.830	0.890	0.811	0.966	0.838	0.875	0.800	0.945	0.832

## Data Availability

We used the publicly available Liver Tumor Segmentation (LiTS) dataset [17] and our private dataset. The private dataset is not publicly available due to privacy and ethical reasons.

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
