# Peer review of "A Boundary-Enhanced Liver Segmentation Network for Multi-Phase CT Images with Unsupervised Domain Adaptation"

_bioengineering, 2023, doi:10.3390/bioengineering10080899_

Round 1

Reviewer 1 Report

The paper proposes a novel methodology: a boundary-enhanced liver segmentation network for multi-phase CT images with unsupervised domain adaptation. Overall, the paper is well written but needs some improvements to enhance the reader's understanding.

1.      Please mention the quantitative results in the abstract. 

2. Separate the related work from the introduction section. 

3. Make a comparison table highlighting the strengths and weaknesses of previous and proposed models.

3. More explanation about the objective function is required, as to how you calculate the values of different losses and other parameters used in equation 8.

4.      Include the details about the trainable parameters in your proposed network as compared to other SOTA methods. 

Author Response

Dear Sir/Madam,

Thank you.

Reviewer 2 Report

Dear authors,

The paper is fine and the results presented are appropriate.

I would like to mention that, if the intermediate results like, training accuracy, loss along with a sample segmentation result is implemented, it will be fine to understand the work executed.  (Please include the intermediate results as in the work: https://doi.org/10.1016/j.procs.2023.01.250 - no need to include in reference, the currenr reference is fine).

If possible, include some other performance metrics along with DC, IoU and TRP.

Author Response

Dear Sir/Madam,

Thank you.

Reviewer 3 Report

1.     Your Citing References are in the wrong order.  Note: Citing References in the main text should be arranged in order from a number [1]. 

2.     Related work is considered essential; thus, it is required to add a section for related work, presenting the most recent studies.

3.     The research gap should be presented based on the limitations of the previous works. 

4.     The dataset is considered as poor contrast; thus, it is required to explain in more detail how this challenge has been solved based on the proposed model.

5.     I recommend to add the architecture and parameters of the proposed model in a Table. 

6.     How the proposed model has been trained?

7.     Many paragraphs in the manuscript are missing citations, thus, it is required to provide appropriate references. 

8.     In section 3 it is mentioned that the current article has performed preprocessing stage, however, it is not mentioned in the main text. It is required to add a section for the preprocessing stage. 

9.     More results should be provided for the proposed model based on the Jaccard index, recall, and accuracy. 

10.  I see the discussion is too short. Optionally, you can input some more things to discuss.

11.  What are the limitations of the enhanced U-Net model?

Author Response

Dear Sir/Madam,

Thank you.

Round 2

Reviewer 1 Report

Most of my comments are addressed. I recommend another minor improvement.

- In the comparison table include the strengths and weaknesses of the proposed method along with the previous methods. 

Author Response

Dear Sir/Madam,

Thank you

Sincerely
